# Integrated analyses of murine breast cancer models reveal critical parallels with human disease

Jonathan P. Rennhack[1], Briana To[1], Matthew Swiatnicki[1,2], Caleb Dulak[1], Martin P. Ogrodzinski[1,3], Yueqi Zhang[1], Caralynn Li[1], Evan Bylett[1], Christina Ross[4], Karol Szczepanek[4], William Hanrahan[1], Muthu Jayatissa[1], Sophia Y. Lunt [3,4], Kent Hunter [5] & Eran R. Andrechek[1]

Mouse models have an essential role in cancer research, yet little is known about how various models resemble human cancer at a genomic level. Here, we complete whole genome sequencing and transcriptome profiling of two widely used mouse models of breast cancer, MMTV-Neu and MMTV-PyMT. Through integrative in vitro and in vivo studies, we identify copy number alterations in key extracellular matrix proteins including collagen 1 type 1 alpha 1 (COL1A1) and chondroadherin (CHAD) that drive metastasis in these mouse models. In addition to copy number alterations, we observe a propensity of the tumors to modulate tyrosine kinase-mediated signaling through mutation of phosphatases such as PTPRH in the MMTV-PyMT mouse model. Mutation in *PTPRH* leads to increased phospho-EGFR levels and decreased latency. These findings underscore the importance of understanding the complete genomic landscape of a mouse model and illustrate the utility this has in understanding human cancers.

---

[1] Department of Physiology, Michigan State University, East Lansing, MI 48824, USA. [2] Department of Microbiology and Molecular Genetics, Michigan State University, East Lansing, MI 48824, USA. [3] Department of Biochemistry and Molecular Biology, Michigan State University, East Lansing, MI 48824, USA. [4] Department of Chemical Engineering and Materials Science, Michigan State University, East Lansing, MI 48824, USA. [5] Laboratory of Cancer Biology and Genetics, National Cancer Institute, National Institutes of Health, Bethesda, MD 20892, USA. Correspondence and requests for materials should be addressed to E.R.A. (email: andrech1@msu.edu)

Breast cancer is an extremely prevalent disease, with an estimated one in eight women experiencing the disease in her life. Owing to its prevalence, it represents a huge public health concern. A hallmark of the breast cancer is its heterogeneity. Heterogeneity in human breast cancer is present in genomic events, gene expression, metastatic potential, and treatment response. Clinically, breast cancer has been classified on the status of estrogen receptor (ER), progesterone receptor (PR), human epidermal growth factor receptor 2 (HER2), as well as proliferative markers such as Ki67. However, with the advancement of transcriptomics and the advent of DNA microarray technology breast cancers began to be profiled on their global genomic profile. Based upon hierarchical clustering of breast cancer patients five subtypes were identified Luminal A, Luminal B, Basal, HER2 positive, and normal-like[1,2]. To identify heterogeneity of the disease from a multi-omic perspective, international efforts have been undertaken to profile breast cancer using multiple -omic platforms[3,4]. These efforts have been productive in identifying a number of genomic events present in breast cancer patients.

To assess gene function in tumor biology, studies have used model systems, including genetically engineered mouse models (GEMMs). However, the genetic landscape of breast cancer mouse models is poorly understood. Recent studies in lung and skin cancers have characterized mouse models of those diseases and have drawn important conclusions about each model[5,6].

The transcriptional landscape of breast cancer GEMMs, with particular attention paid to relationships with human breast cancer has been extensively studied[7–9]. As follow up to these studies the copy number landscape of the models has been predicted[10,11]. Here, we present whole-genome sequencing data of two highly utilized mouse models of breast cancer, MMTV-Neu[12] and MMTV-PyMT[13]. In PyMT tumors we identify a highly conserved mutation in the protein tyrosine phosphatase receptor (Ptprh) resulting in elevated EGFR activity and erlotinib sensitivity. In Neu tumors, a copy number alteration including collagen type 1 alpha 1 (Col1a1) and chondroadherin (Chad) alters metastatic potential, which is validated through genetic ablation. Altogether, this data demonstrates that genomic alterations beyond the initiating oncogene need to be considered when choosing a model system for breast cancer.

## Results

### Transcriptional characterization of mouse model.
To characterize the genomic landscape of the MMTV-Neu and MMTV-PyMT tumors, we created a tumor database with complete phenotypic characterization, including tumor latency, histology, and metastatic burden (Supplementary Table 1). Representative tumors from this database were selected for whole-genome sequencing and whole-transcriptome profiling by microarray. The analysis pipeline then correlated phenotypic changes with molecular profiling, including transcriptomics and sequence alterations. The resulting genes were then filtered through human breast cancer datasets to ensure relevance to human breast cancer and confirmed with in vitro/in vivo experiments (Fig. 1a). To combine two separate batches of MMTV-PyMT and MMTV-Neu we used Bayesian Factor Regression Methods (BFRM) analysis and visualized with principal component analysis (PCA) (Supplementary Fig. 1). A high degree of transcriptomic diversity both between and within each model was observed in hierarchical clustering (Fig. 1b). Consistent with previous studies, there are similarities in the signaling pathways of MMTV-Neu and MMTV-PyMT models[14]. We saw very little difference in the transcriptional landscape of the models. To minimize the bias potentially introduced by clustering approaches we also used PCA to visualize the transcriptome of the models (Supplementary Fig. 2). As expected, these results demonstrate that heterogeneity correlated with tumor histological subtype rather than tumor model, consistent with recent studies[7,15].

To identify on a transcriptional level if the mouse models represented human breast cancer tumors, we co-clustered the mouse tumors with TCGA tumors from each molecular subtype (Supplementary Fig. 3). Consistent with previously published data, we observed that the considerable diversity within the mouse model led to a cohort of tumors, which co-clustered with the human disease and a cohort that has a unique transcriptional profile. We specifically identified a cohort of MMTV-PyMT tumors that co-cluster with basal tumors as well as a significant portion of MMTV-Neu and MMTV-PyMT tumors, which have transcriptional profiles that are similar to HER2 + breast cancer. It was hypothesized that these transcriptional differences were driven by genomic changes.

### Genomic sequencing of mouse models.
Following standard informatic pipelines, the whole-genome sequence was analyzed. In mapping, the location of Neu and PyMT insertion was uncovered. MMTV-Neu is inserted into an intron of FARS2 (Phenylalanyl-tRNA synthetase, mitochondrial) on chromosome 13 and PyMT is in an intron of a RIKEN protein on chromosome 8. To validate bioinformatic calls of single-nucleotide variants (SNVs) and copy number variants (CNVs) we used PCR and quantitative PCR (qPCR), observing a validation rate of 85% (Supplementary Table 2). Whole-genome sequencing revealed large differences in the genomic landscape of the MMTV-Neu (Fig. 1c) and MMTV-PyMT (Fig. 1d) tumors. The two tumor models had similar numbers of SNVs (Fig. 1e and Supplementary Table 3). However, both models were ~20x more stable than human breast tumors with 0.049 mutations/megabase in the mouse models in comparison to an average of ~1 mutation/megabase in breast cancer[16]. Copy number alterations (Fig. 1f and Supplementary Table 4) and translocations (Fig. 1g and Supplementary Table 5) were more frequent in the MMTV-Neu model relative to MMTV-PyMT.

To identify consistent alterations that were unique to each model or shared across both, we first identified those genes effected by SNVs, CNVs, and translocations that were present in each model (Fig. 2a). Interestingly, this resulted in identification of a number of key signaling proteins, including those within the PI3K, KRAS, and MAPK signaling cascade. In addition to the initiating oncogene, both mouse models were largely seen to be associated with a number of copy number alterations. To test whether these copy number alterations were consistently targeting similar pathways and processes we utilized an over-representation analysis (Fig. 2b). This resulted in a number of potentially interesting gene signaling pathways, including consistent alterations to PI3K/AKT/MTOR signaling in the MMTV-Neu model and E2F1 in the MMTV-PyMT tumors. Each of these models have been shown in the literature to have high signaling in those respective pathways[14] and the CNV data may show a potential mechanism for upregulation of these signaling pathway. Interestingly, we noted that both models showed consistent copy number alterations in the MAPK signaling cascade with each individual tumor containing multiple nodes amplified (Fig. 2c).

### Copy number alterations in mouse models.
Owing to the fact genomic events in each mouse model were dominated by copy number alterations as compared to single-nucleotide changes or translocations, we wanted to compare the mouse models to human HER2 + breast cancer, which is also dominated by copy

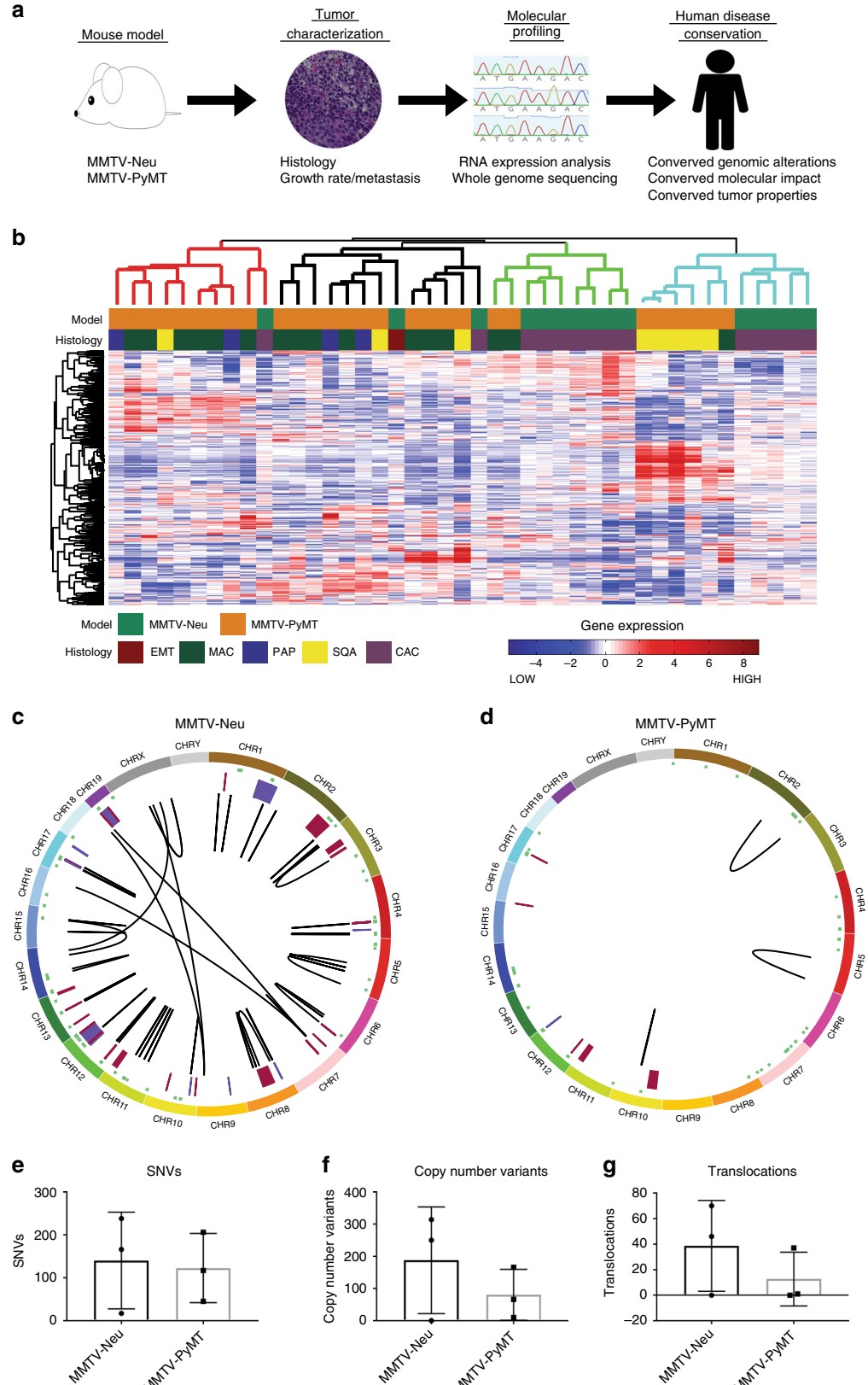

number alterations. Interestingly we saw overlap of partial genomic events but not necessarily the entire event when comparing mouse models to the human tumors. An example of this is the chromosome 12p amplification in humans, which has been shown to contain the *KDM5A* amplification as a driver gene. This event was shown to be partially conserved in the mouse model, which shows a large amplification event on chromosome 6 also containing the *KDM5A* locus (Fig. 3a).

To identify events conserved between mouse models and human breast cancer, we compared amplified and/or deleted

**Fig. 1** Genomic landscape of MMTV-Neu and MMTV-PyMT tumors. The schematic representation of the project workflow is depicted (**a**), where mammary tumors from two major mouse models are completely characterized through histological, molecular, genomic, and transcriptomic methods. After data integration and analysis, the tumors were compared to human cancers at both genomic and phenotypic levels. Gene expression patterns from MMTV-Neu and MMTV-PyMT tumors were compared by unsupervised clustering, revealing substantial heterogeneity both between and within models. Tumors clustered largely based on histological subtype and not simply genotype. SQU–squamous, MAC–microacinar, PAP–papillary, and CAC–comedo-adenocarcinoma (n = 15 for MMTV-Neu, n = 25 for MMTV-PyMT) (**b**). Circos plots from whole-genome sequencing results for MMTV-Neu (**c**) and MMTV-PyMT (**d**) tumors revealed differences between the strains for genomic alterations. Plots display from outside in; Chromosomal location (Each chromosome is unique color), SNVs (green), copy number alterations (Amplification–Red and Deletions–Blue), and translocations (black lines). Variation from multiple tumors is shown for single-nucleotide variants (**e**), copy number variants (**f**), and translocations (**g**) (n = 3 for each). All error bars present are standard deviation

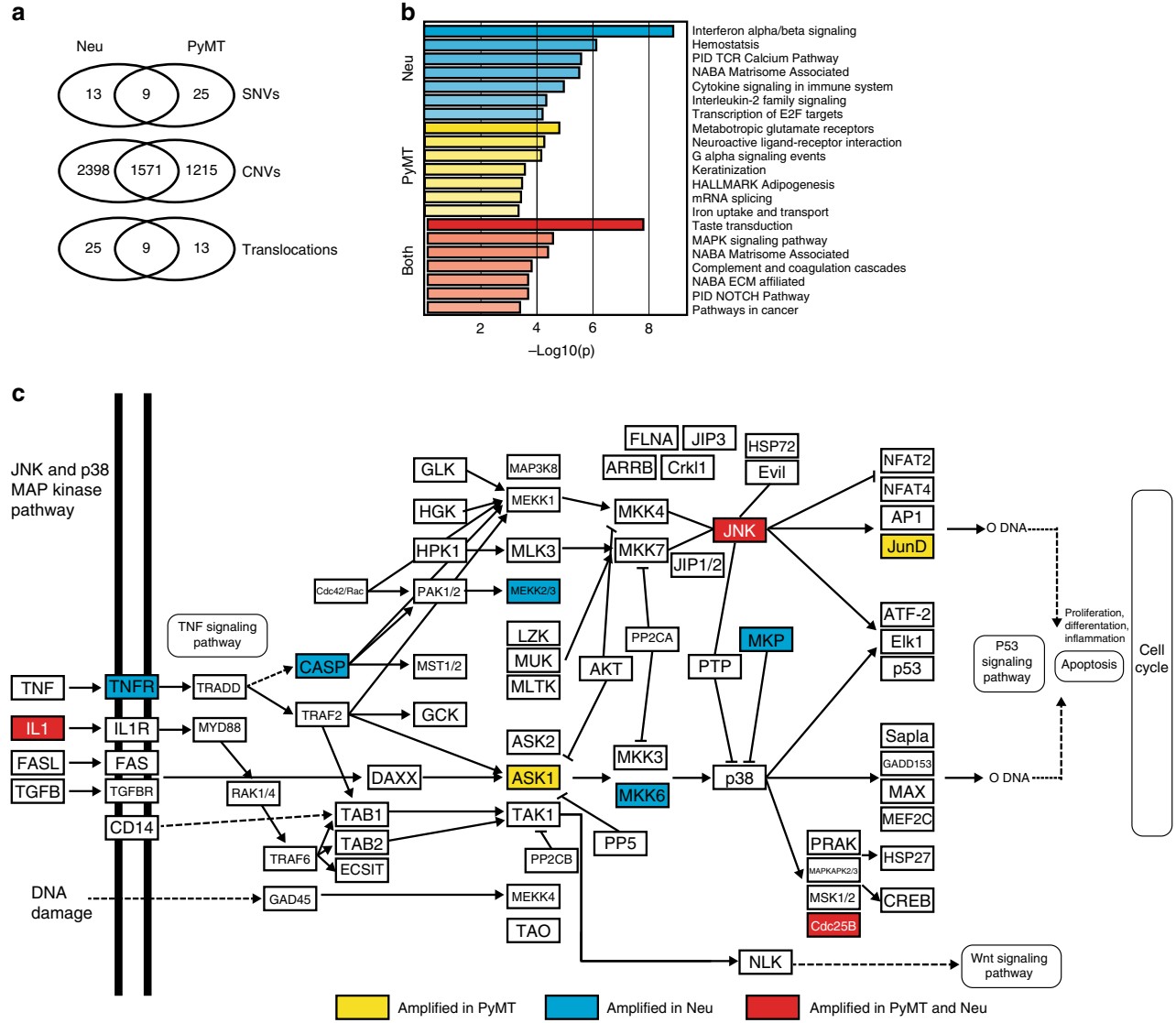

**Fig. 2** Signaling pathways are differentially altered in MMTV-Neu and MMTV-PyMT. Key differences in the alterations found in the MMTV-Neu and MMTV-PyMT (**a**) are apparent in the presence of SNVs (top), CNVs (middle), and translocations (bottom). The presence of copy number variants were apparent in both models. We used over-representation analysis through the assistance of metascape to identify consistently altered genesets (**b**) and there were important genesets implicated in tumor progression in the copy number profiles of the MMTV-Neu (top), MMTV-PyMT (middle), and both models (bottom). One key gene set over-represented is the MAPK signaling cascade (**c**) where key signaling nodes can be shown to be altered in each model

genes in each mouse model to those that were amplified or deleted in HER2 + cancer as identified by the TCGA group (Fig. 3b). While there was significant overlap in both models and the human disease it was surprising that the MMTV-PyMT had a much higher number of amplification and deletion events consistent with the HER2 + samples than the MMTV-Neu

samples. To continue to explore the copy number landscape of the MMTV-Neu tumors we filtered to those overlapping with human disease. Two out of the three mouse tumors sequenced did show amplification of driver genes (Fig. 3c). However, these were not consistent across multiple tumors and may not be functionally relevant.

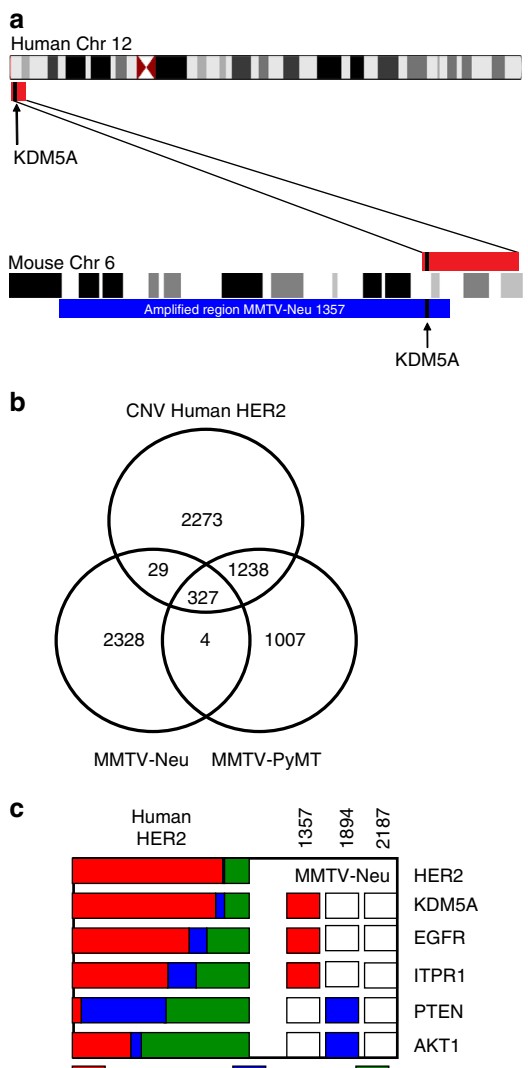

**Fig. 3** Copy number profiles of mouse models show similarities with human HER + tumors. Portions of many amplicons are shared between the mouse models and human breast cancer, including the *KDM5A* amplicon (**a**). The human amplicon (red, including *KDM5A* in black) on chromosome 12 consists of numerous genes that are found in the mouse on chromosome 6 (red). The large amplification event in Neu tumors is shown in blue and both amplicons contain *KDM5A* (**a**). Consistent overlap between the mouse models and HER2 + human breast cancers was noted (**b**) with there being more overlap between the MMTV-PyMT and HER2 + model. To determine if these overlapping genes were driving genes, we identified those genes, which were amplified or deleted in the MMTV-Neu tumors and called as driving genes in the TCGA HER2 + cohort (**c**). CNV (red, amplified and blue, deleted) and mutation status (green) is noted for both human and mouse tumors

**Identification of 11D amplification in MMTV-Neu**. To find copy number alteration that are functionally relevant we identified genes, which were consistently amplified or deleted in the mouse models, as well as in human disease. This analysis identified 11 candidate genes, which were highly altered in breast cancer (Supplementary Fig. 4) and predicted to impact tumor biology based upon a literature screen. qPCR gene copy number analysis across an extended tumor panel (15 MMTV-PyMT, 10 MMTV-Neu) identified the rate at which each copy number variant occurred throughout the model (Fig. 4a). This analysis showed that while each of the copy number variants predicted

through bioinformatic approaches were valid (Table 2S), the depth of the amplification was largely around 1.5-fold indicating shallow amplification events (Fig. 4b). Interestingly, we identified the largest diversity of copy number profiles in the 11D locus. This locus includes a total of 40 genes, 19 with transcriptomic differences. Depending on the presence or absence of the locus, the tumors exhibited striking differences in structure and behavior. We identified dramatic differences in the tumors with the presence of an 11D amplification with regards to collagen content through a Masson's trichrome (Fig. 4c) stain and the presence of metastatic lesions in the lungs (Fig. 4d).

To identify the driving genes of the metastatic phenotype associated with 11D amplification, we examined human breast cancer for distant metastasis free survival outcomes. We then created CRISPR-Cas9 generated knockouts of two potential metastasis related proteins within the region, Collagen type 1 alpha 1 (Col1a1) and Chondroadherin (Chad). Knockouts were generated in two mouse driven tumor cell lines NDL2–5[17] and PyMT 419[18] (Supplementary Fig. 5) and compared to multiple cas9-transfected clones with no sgRNA. NDL2–5 is an 11D amplified Neu driven line, while the 419 line is diploid for the 11D locus and is driven by PyMT expression. Importantly, we saw no difference in growth rate with the loss of Col1a1 or Chad in our systems (Supplementary Fig. 6). Knockouts of each gene in both cell lines revealed defects in the ability to migrate in a wound-healing assay (Fig. 4e, f) when compared to the non-edited 419 and NDL wild-type clones. Migration was partially rescued with addback of wild-type Col1a1 or Chad, demonstrating that migration defects were not due to off-target effects (Supplementary Fig. 7). Defects in lung colonization in a tail vein injection were also observed with the Col1a1 and Chad knockout cell lines (Fig. 4g, h).

**Conservation of Col1a1/CHAD amplification in breast cancer**. Mouse chromosome 11D is conserved in humans and is analogous to chromosomal region 17q21.33. There is similar amplification event at 17q21.33, including *COL1A1* and *CHAD* that occurs in 8% of breast cancer patients. Array CGH from the TCGA data[3] demonstrates that *COL1A1/CHAD* amplification was distinct from HER2 amplification (Fig. 5a). Importantly, this amplification is subtype specific; 25% of Her2 + breast cancers have a co-amplification of the 17q21.33 region along with the HER2 amplicon while only 6% of Luminal A, 7% of Luminal B, and 1.2% of Basal breast cancers have amplification (Fig. 5b). To investigate the transcriptional impact of the amplification event, we used weighted gene correlation network analysis[19]. This identified a robust transcriptional signature that differentiated *COL1A1/CHAD*-amplified, Her2-positive tumors from Her2-positive tumors without the amplification event (Supplementary Table 6). Unsupervised hierarchical clustering readily identified separation of the two HER2-positive subtypes based on this signature (Fig. 5c). These correlated genes were used in a predictive signature to correlate patient outcome with predictive amplification status (Supplementary Fig. 8), revealing that metastasis was associated with the amplification event (Fig. 5d).

To test whether COL1A1 and CHAD were driving the metastasis phenotype in human breast cancer, we used CRISPRi[20] to create a pooled population of COL1A1 and CHAD knockdown in the Her2 amplified, *COL1A1/CHAD* amplified breast cancer line BT-474. These knockdowns were shown to have a reduction in CHAD (Supplementary Fig. 5G) and Col1a1 (Supplementary Fig. 5F), respectively, as well as the inability to create collagen fibers as identified through a Masson's trichrome (Supplementary Fig. 5G). While we saw no differences in the proliferation rate of these lines in vitro, these knockdowns

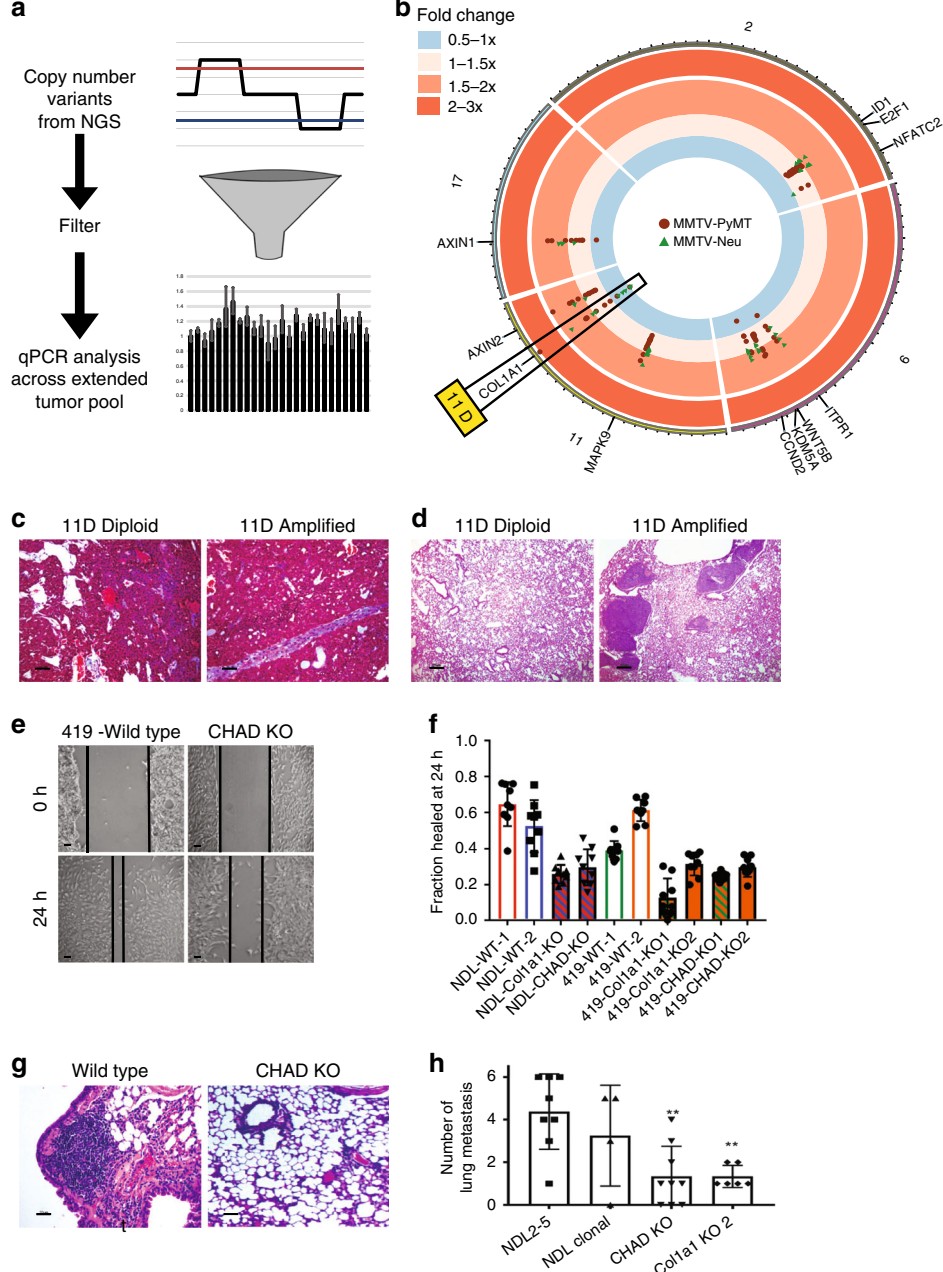

**Fig. 4** Copy number alterations alter metastatic potential. Schematic representation of filtering of copy number variants (**a**) where copy number variants were detected from NGS and filtered to the top 11 variants by selecting those genes that encompassed both mouse models and were conserved in human breast cancer. These genes were then assayed using qPCR analysis across a larger panel of MMTV-Neu ($n = 10$, green triangles) and MMTV-PyMT ($n = 15$, red circles) tumors and depicted using a circos plot for genes in chromosomes 2, 6, 11, and 17 (**b**). Heterozygous deletion on the interior of the plot to a threefold amplification on the exterior of the plot is shown. A key copy number alteration in the 11D region encompassing the *Col1a1* gene (boxed) was observed to correlate with reduction and lack of collagen alignment in Masson's trichrome staining (**c**) and an increase in metastases in the lungs of mice with *Col1a1* amplification in the primary tumors (**d**). CRISPR-Cas9-mediated knockout of two key genes within this region, *Col1a1* and *Chad*, show defects in wound healing (**e**, **f**) (scale bar = 100 μm) ($n = 9$) (Colors denote significance by students two-tailed, unpaired $t$-test, $P < 0.01$, vs. the same colored control). Knockout also impaired the ability to colonize the lung through a tail vein injection (**g**, **h**) (NDL2–5 $n = 12$, NDL2–5 Clonal $n = 4$, Chad KO $n = 9$, Col1a1 KO $n = 6$). (** = $P < 0.01$, students two-tailed, unpaired $t$-test, for **h**). Scale bar of 20 μm displayed on histological image for reference. All error bars denote standard deviation

showed a decreased ability to migrate in a wound-healing assay (Fig. 5e, f). When injected into the mammary fat pad we saw no difference between the WT BT-474 ddcas9 population and the COL1A1 or CHAD knockdown populations in their ability to form tumors. This take rate was consistent with previously published rates[21]. We also saw no difference in growth rates with wild-type and knockdown population (Supplementary Fig. 6B) consistent with the in vitro growth rate data. Importantly, the knockdown lines were unable to metastasize to the lung after being injected into the mammary fat pad (Fig. 5g, h). Altogether these data underscore the importance of identifying copy number variation in mouse models of cancer.

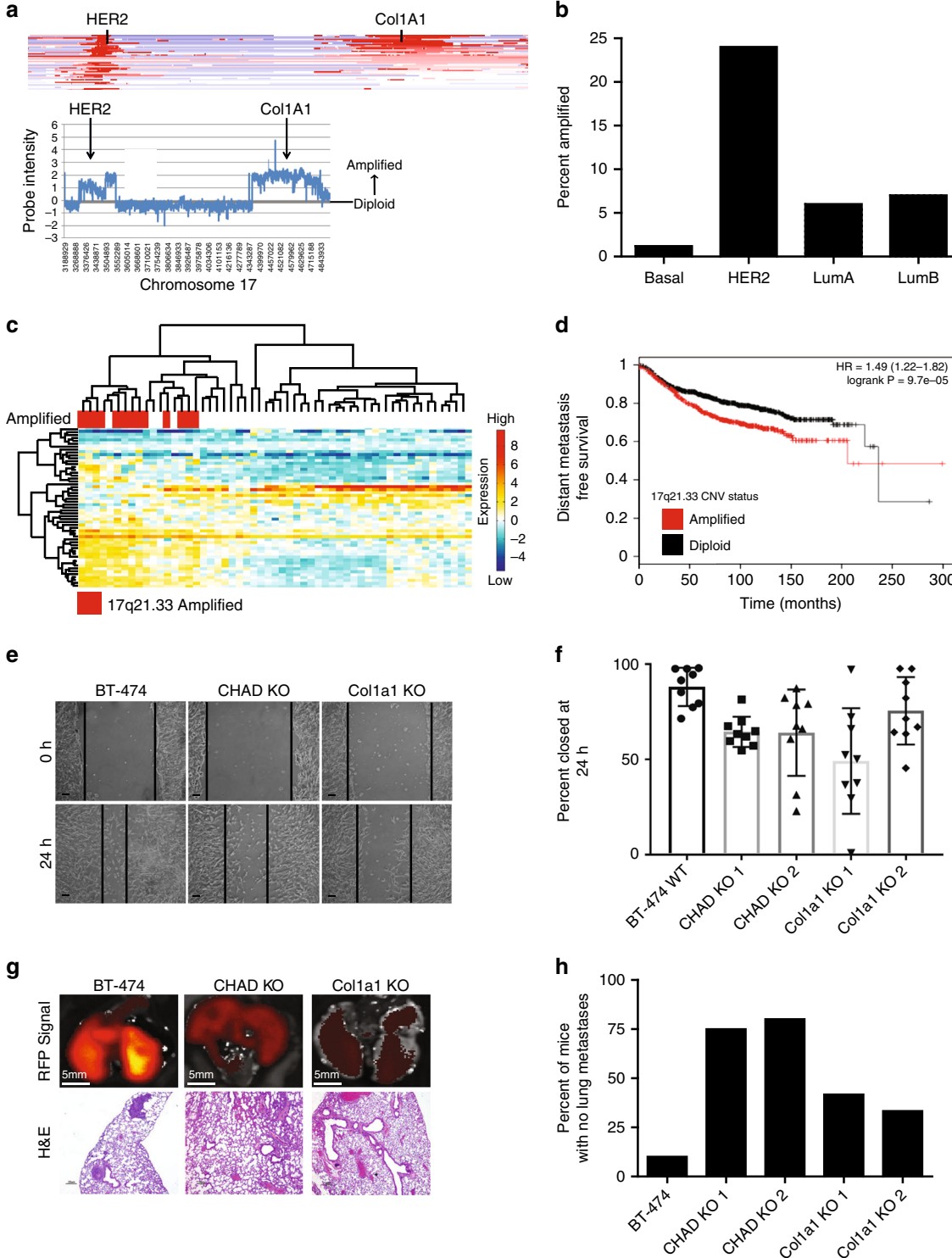

**Fig. 5** 11D amplicon presence and function is conserved in human breast cancer. TCGA breast cancer copy number dataset analysis revealed co-amplification of *HER2* and the *COL1A1* locus through a heatmap across chromosome 17 of multiple samples with each row representing an independent patient sample (**a**-top) with red representing amplification and blue representing deletion. The *COL1A1* amplification event occurred independently of HER2 (A-bottom) as identified by probe intensity of aCGH data of a single TCGA breast cancer patient. The *COL1A1/CHAD* amplification event was disproportionately found in HER2-positive tumors and is present in ~25% of Her2-positive tumors (**b**). Gene expression of HER2 + samples with and without the 17q21.33 amplification demonstrated a unique gene expression profile as identified by unsupervised hierarchical clustering (**c**) and significantly worse overall survival within the KMplotter dataset ($P < 0.001$, log rank test) (**d**). CRISPRi-mediated knockdown of CHAD and COL1A1 in human cell line BT-474 resulted in defects in wound healing (**e, f**) (* = $P < 0.05$, students two-tailed, unpaired *t*-test, $n = 9$) and distant metastasis to the lung after orthotopic injections (**g, h** $n = 10$, for WT, $n = 4$ for CHAD KO1, $n = 5$ for CHAD KO2 $n = 12$ for Col1a1 KO1, $n = 6$ for Col1a1 KO2). Scale bars of 10 μm (**e**), 5 mm (**g** top), and 20 μm (**g** bottom) displayed on image for reference. All error bars presented are standard deviation

**Mutational landscape of mouse models**. In addition to copy number alterations, the whole-genome sequence data resulted in the identification of numerous mutations (Fig. 1c–e). When Catalog Of Somatic Mutations In Cancer (COSMIC) mutational signatures[22,23] were applied to the models, it was observed that the tumor models had similar mutational processes (Supplementary Fig. 9). The MMTV-Neu and MMTV-PyMT tumors both contain the same trinucleotide context of their mutation spectrum. The mutation spectrum shows all nucleotide substitutions present with a slight bias towards C/T and T/C transitions. When compared to the human mutational signatures, the mutational processes present in both mouse models closely resembles COSMIC signature 5 (Fig S9C). This signature has been shown to be present in breast cancer patients with disease associated with late onset[24], indicating a similar mutational process in both the human disease and mouse models

Distribution of SNVs reflected patterns seen in the transcriptional data (Fig. 1b) with some events shared between Neu and PyMT tumors while others were unique to the models. Considerable SNV diversity within a model was also prevalent. For instance, the MMTV-Neu model had no genes with shared mutations in all samples and only five genes containing a coding, non-synonymous mutation in more than one sample (Supplementary Fig. 10). Notably we identified mutations within Mucin 4 (*Muc4*), which are potentially impactful due to *Muc4*'s emerging roles in Her2-positive cancer and metastasis[25]. Interestingly, we observed that PyMT-induced tumors had more SNVs in the coding regions of the genome. Specifically, these mapped to 34 genes, 9 of which overlapped with Neu tumors. A number of genes with coding mutations specifically in PyMT tumors, including *Matn2, Plekhm1, Muc6*, and *Ptprh* were observed. Matn2[26], Plekhm1[27], and Muc6[28] have all been demonstrated to have roles in tumor progression and metastasis and may contribute to the high metastatic capacity of the MMTV-PyMT model.

To test the frequency of these coding mutations in the models as a whole, we selected a population of 10 MMTV-Neu tumors and 15 MMTV-PyMT tumors for targeted resequencing. From these tumors we extracted genomic DNA and performed PCR-based amplification followed by Sanger sequencing of *Matn2*, *Plekhm1*, and *Ptprh*. While *Matn2* and *Plekhm1* confirmed the whole-genome sequencing variant calls in the sequenced tumors, additional mutations were not found.

**Identification and characterization of *Ptprh* mutation**. Strikingly, *Ptprh* was found to be mutated in 81% of MMTV-PyMT tumors. Furthermore, the *Ptprh* mutation was shown to be homozygously mutated in 21% of PyMT tumors and heterozygously mutated in 60% of PyMT tumors (Fig. 6a, b). Surprisingly, an identical C to T mutation was observed in each tumor resulting in a valine residue being converted to a methionine at amino acid 483 (V483M). To test for the conservation of mutations of *Ptprh* in mouse strains beyond FVB/NJ, we sequenced *Ptprh* of MMTV-PyMT models in a C57/Bl6, C57/Bl10, CAST, and MOLF backgrounds, as well as a different inbred MMTV-PyMT FVB/NJ line. This analysis showed consistent mutation in the structural fibronectin domains (FN3) and the phosphatase domain of Ptprh (Fig. 6c). Interestingly, we found that the two FVB models contained different mutational patterns, indicating an impact of environmental and potential epigenetic causes of mutational hotspots.

Given that recent work identified the target of PTPRH as EGFR[29], we hypothesized that EGFR was not dephosphorylated with *Ptprh* mutation. Testing this, we observed that the V483M mutation correlated with pEGFR levels (Fig. 6d, e). With the resulting increase in EGFR activity, we also observed a significant decrease in tumor latency (Fig. 6f). With an increase in EGFR activity, it was possible that tumors with mutant *Ptprh* would be dependent upon EGFR signaling. To test this prediction, cell lines derived from *Ptprh* wild-type and mutant tumors were treated with EGFR targeted therapy. After 48 h, tumors containing *Ptprh* mutations were shown to be more sensitive to erlotinib treatment (Fig. 6g).

Given the role of EGFR in lung cancer, we next sought to determine if there was a non-EGFR mutant patient population within lung cancer that could benefit from EGFR inhibition. Examination of the pan-lung TCGA data revealed 5% of patients with a mutation in *PTPRH*. Importantly, these mutations were shown to be mutually exclusive from EGFR, indicating that patients were likely not treated with EGFR tyrosine kinase inhibitors. To confirm the impact of *PTPRH* mutations on EGFR activity in human lung tumors we used gene set enrichment analysis to predict EGFR activity of each mutant *PTPRH* sample. This analysis revealed four key hotspots of mutations driving high EGFR activity, including three in the FN3 domains and one in the phosphatase domain of *PTPRH* (Fig. 6i).

## Discussion

Altogether, these data emphasize the heterogeneity within tumor models and the importance of understanding the genomic landscape within the tumors. Here, we presented a proof-of-concept study to identify a number of events that have influenced key tumor phenotypes, including metastasis and tumor latency. Through the analysis we have also been able to identify similarities between the mouse models and the human disease from an SNV, CNV, and translocation standpoint. Despite the limitations with the number of samples, this study offers a unique opportunity to identify genomic alterations which impact tumor behavior and treatment response. These findings have direct therapeutic impact with a potential impact on patient therapeutic intervention and metastatic progression of their disease and underscore the important role genetically engineered mouse models have in understanding tumor biology.

The presence of the *Col1A1/CHAD* amplification event in the mouse model mirrors the 25% of human HER2 + ve breast cancers that also had amplification of a structurally conserved region. Given the potential role for these genes in metastasis, this work clearly indicates that the MMTV-Neu system is an appropriate model for select facets of HER2 tumor biology, including the additional amplification event. However, the lack of amplification of genes surrounding *erbB2* in the mouse model indicates that other models with *erbB2* amplification[30] may be more suitable for other studies.

In the PyMT model system, tumor onset is rapid with mice developing tumors 45 days after birth. Despite this rapid onset, the data presented here indicates that over 80% of PyMT tumors acquire the identical mutation in *Ptprh*, suggesting that there is a significant evolutionary pressure applied during the initial transformation. Given that the *Ptprh* mutant does not dephosphorylate EGFR, this results in unchecked activation of a key signaling pathway. In part, this event may impact metastatic progression of this model system. The most likely impact from this discovery will be in the identification of additional tumor patients that may benefit from EGFR TKI therapy.

Taken together, this manuscript provides a resource for investigators to determine how well the subtype of cancer they examine is represented by MMTV-Neu and MMTV-PyMT mouse model systems. While we have explored two genomic events, others of interest are noted and their impact will be elucidated. The data from these two models also underscores the importance of a complete characterization of GEMMs for human cancer.

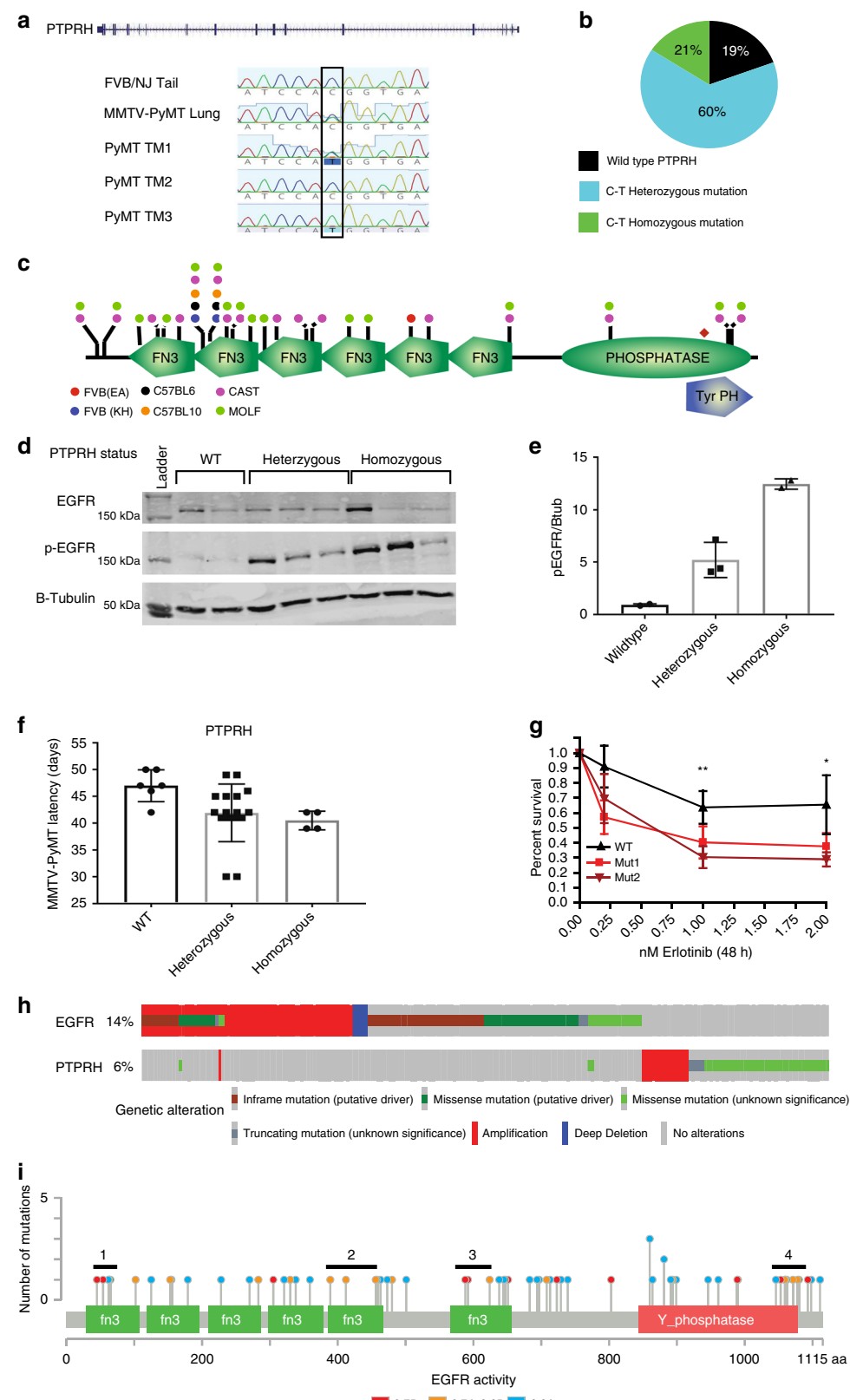

## Methods

**Animal studies**. All animal husbandry and use was conducted according to local, national, and institutional guidelines. The study received ethical approval from the Michigan State University Institutional Animal Care & Use Committee (IACUC) under AUF 06/18–084–00. The MMTV-Neu[12] and MMTV-PyMT[13] mice were in the FVB background. MMTV-PyMT634 and MMTV-Neu mice were obtained from The Jackson Laboratory. Mice were monitored twice weekly for tumor initiation and growth. At a 2000 mm[3] endpoint, mice were necropsied. For mice with multiple tumors the endpoint was established when the primary tumor was at 2000 mm[3]. Tumors and lungs were collected for genomic analysis, hematoxylin and eosin staining for histological subtyping and presence of pulmonary metastases. The number of metastasis was quantified using a single cut through the lung and count of the number of micro-metastases in that plane. Masson's trichrome staining was used to examine tumors for collagen deposition using standard methods.

**Fig. 6** *PTPRH* mutations are conserved in MMTV-PyMT and human lung cancer. Phosphotyrosine receptor, *Ptprh* was shown through Sanger sequencing to have C-T mutations in the MMTV-PyMT tumors while tail and lung samples were normal (**a**). Expanding targeted sequencing to additional samples revealed a conserved heterozygous mutation in 60% and homozygous mutation in 21% (**b**) of MMTV-PyMT tumors ($n = 45$). Sequencing revealed multiple mouse backgrounds (FVB, C57BL6, C57BL10, CAST, and MOLF) have varied mutations clustered in the functional domains of the *Ptprh* proteins (**c**). Increases in pEGFR signaling (**d**, **e**) and a decrease in tumor latency (F, * = $P < 0.05$, students two-tailed, unpaired *t*-test) was correlated with the mutant *Ptprh* allele (V483M) within the FVB background. Cell lines derived from *Ptprh* mutant (V483M) PyMT tumors showed an increased response to EGFR targeted therapy including erlotinib (**g**, $n = 3$, ** = $P < 0.01$, * = $P < 0.05$, students two-tailed, unpaired *t*-test). In human lung cancer (TCGA-pan-lung cancer) ~5% of patients have a mutation in PTPRH, which are mutually exclusive from EGFR (**h**). High EGFR activity, as determined by gene set enrichment analysis, is associated with mutations clustered within the four indicated structural and functional domains of PTPRH as seen by the colors in the lollipop plot for EGFR activity (**i**). All error bars presented are standard deviation

**Whole-genome sequencing**. Flash frozen tumor pieces were ground, and DNA was extracted with the Qiagen Genomic-tip 20/G with the manufacturer's protocol. DNA was sequenced to a depth of 40x with paired end 150 base pair reads on an Illumina HiSeq 2500 using the Illumina TruSeq Nano DNA library preparation.

**Transcriptomic profiling**. Transcriptome data for this study was previously published[7,31,32]. Affymetrix expression console was used to normalize each individual dataset using RMA normalization. To remove batch effects between datasets BRFM normalization[33] was performed with standard parameters and visualized with PCA.

**Clustering**. Unsupervised hierarchical clustering was performed using Cluster 3.0 and the Broad institute's Morpheus interface. Heatmaps were created using the MATLAB imagesc function.

**Variant calling**. Generated.fastq files were assessed for quality control using FASTQC analysis (http://www.bioinformatics.babraham.ac.uk/projects/fastqc). Reads were trimmed for quality using Trimmomatic[34]. After trimming, data was reassessed for quality using FASTQC. Then reads were aligned to the mm10 mouse reference genome using BWA-mem[35]. After alignment, base recalibration and pcr-induced biases were removed using PICARD tools (http://broadinstitute.github.io/picard). For variant calling we utilized four software packages, GATK[36], Mutect2[37], Strelka[38], and SomaticSniper[39]. To be a legitimate variant we filtered to only those variants called by 3 of the 4 packages. To control for differences in the FVB strain and the mm10 reference genome we used previously published normal FVB tissue (ERR046395)[40]. To call copy number and structural variants we used Delly[41]. For copy number we used default quality control settings and only analyzed those copy number events, which had precise boundaries and were larger than 100 KB. For translocations we used default quality control setting and precise breakpoints.

**Variant verification and extended tumor panel sequencing**. For verification of SNVs we used PCR (primers in Supplementary Data)-based amplification followed by Sanger sequencing. For validation of CNVs, we used qPCR (primers in Supplementary Data) on the genomic DNA with the Quantabio PerfeCTa SYBR green kit under the manufacturer's specifications. Primers for PCR and sequencing are listed in Supplementary Data.

**Circos visualization**. Representative MMTV-Neu and MMTV-PyMT samples were chosen to be displayed as CIRCOS[42] plots. CIRCOS plots were generated using CIRCOS v 0.69 and SNVs, CNVs, and translocations were mapped according to their location on the mm10 genome.

**Mutation signatures**. Owing to the low mutational burden of MMTV-Neu and MMTV-PyMT tumors, mutations were combined into a signal analysis for each model. These samples were processed with MutSpec-NMF[43] for trinucleotide context and comparison to the known human mutation signatures.

**Cell lines**. The PyMT 419 cell lines were a gracious gift from Dr. Stuart Sell and Dr. Ian Guess[18]. The NDL2–5 cells lines were obtained as a gift from Dr. Peter Siegel[17]. The BT-474 cell line was obtained from Dr. Kathy Gallo and validated using STR fingerprinting analysis performed at Michigan State University. STR fingerprinting was performed with ProMega Cell ID System run on the ABI 3730xl.

**CRISPR generated knockouts of PyMT 419 and NDL2–5**. CRISPR/Cas9 constructs were created to knockout *Col1a1* and *Chad* in PyMT 419 and NDL2–5. Guides (listed in Supplementary Data) were designed and inserted into Px458, obtained from addgene (Addgene #48138) as a gift from Feng Zhang, using the BBSI insertion site[44]. Cells were sorted using FACS technology into single cells and grown into clonal population, then screened for the presence of INDELs using Sanger sequencing. Knockouts were further confirmed for the NDL2–5 lines using western blot. Guide Sequences are listed in Supplementary Data. Control lines were

transfected with an empty (no sgRNA) and flow cytometry sorted to single-cell clones.

**CRISPRi generated knockdowns in BT-474**. Knockdowns of Col1a1 and CHAD were created in the BT-474 line using CRISPRi technology. Genomic RNA (gRNA) (listed in Supplementary Data) were cloned into a plasmid containing the gRNA under the control of the U6 promoter (Addgene plasmid #60955)[45]. Lenti virus was created for stable expression of this plasmid and the stable expression of KRAB-Cas9 fusion protein (Addgene plasmid #60954)[45]. Cells were infected with KRAB-Cas9 expression virus first and selected for uptake by puromycin treatment. The stable KRAB-Cas9, BT-474 line was then infected with the virus for stable selection of the gRNA for *CHAD* or *COL1A1*. These were then sorted using flow cytometry for RFP expression into a pooled population and validated knockdown through western blot. The populations were also assayed for the ability to create functional collagen fibers through a Mason's trichrome. The plasmids used in the part of the project were obtained through Addgene as a gift from Jonathan Weissman.

**Wound-healing assay**. Wound-healing assays were performed similarly for all cell lines in the manuscript. Cells were grown to 100% confluence in a six well plate then a wound was created in the middle of the plate. Cells were allowed to close the wound for 24 h in the presence of Mitomycin C growth inhibitor then the cells were imaged. Images were quantified for the amount of migration into the wound using ImageJ.

**Tail vein injection**. NDL2–5 *Chad* and *Col1a1* knockout cell lines were injected into the tail vein of syngeneic FVB/NJ mice. Cells were suspended in phosphate buffered saline (PBS) in a single-cell population and injected in a single bolus of $1 \times 10^6$ cells in 50 μL. Mice were monitored for 9 weeks then euthanized. At this point, lungs were collected and stained with Hematoxylin and Eosin to identify the presence of pulmonary metastases.

**Mammary fat pad injection**. NDL2–5 WT and cell lines were suspended in 50 μL PBS and injected into mammary gland number four in syngeneic FVB/NJ mice as a single bolus of $1 \times 10^6$ cells. The mice were monitored twice weekly until tumors reached an endpoint of 2000 mm³.

BT-474 wild-type and CHAD/COL1A1 knockout lines were suspended in a 1:1 concentration of matrigel:PBS mixture (50 μL) and injecting into the mammary gland number four in a single bolus of $1 \times 10^6$ cells. Balb/C nude mice were used for these studies. Tumors were monitored until a size of 1000 mm³. Tumors were then resected, and mice were monitored for an additional 4 weeks. At necropsy lungs were imaged for RFP using the IVIS imaging system and then processed for hematoxylin and eosin staining.

**Human dataset usage**. All human datasets used in this study are publicly available and noted as used in the manuscript. For genomic alteration frequency, the TCGA Breast cancer[3] and the TCGA-pan-Lung cancer[46] datasets were used. For the expression-based survival data the KMPlot.com dataset[47] was used.

**Western blotting**. Western blots in this manuscript were completed under manufacturer's specifications. Blocking was performed for 1 h by incubation at room temperature with the LiCor blocking reagents. Western blots were imaged using the LiCor system. The following antibodies were used: COL1A1 [1:1000] (Origene TA309096), CHAD [1:2500] (Abcam ab104757), EGFR [1:1000] (CST D38B1), pEGFR [1:1000] (Invitrogen PA5–37553), HSP90 [1:1000] (CST 4874S), Beta-tubulin [1:1000] (CST 2128S), anti-rabbit secondary [1:10000] (Licor 926–32211), anti-mouse secondary [1:10,000] (Licor 926–68070).

**Erlotinib sensitivity assay**. Cell lines derived from *Ptprh* mutant and wild-type tumors were seeded at a concentration of 250 cells/mL and subjected to erlotinib treatment for 48 h with the concentrations stated in the manuscript. Eroltinib was purchased from Cayman Chemical. After treatment with erlotinib or DMSO control, cells were given fresh media to grow for 7 days. Cells were then fixed and stained with crystal violet for counting.

**Statistical considerations**. Except where otherwise noted all statistical comparisons are performed with an unpaired students two-tailed, unpaired *t*-test. All error bars presented are standard deviation.

**Reporting summary**. Further information on research design is available in the Nature Research Reporting Summary linked to this article.

## Data availability

The fastq sequencing datasets for the MMTV-Neu and MMTV-PyMT models has been deposited in the NIH sequencing reads archive with the identifier PRJNA541842. The mouse transcriptomic data has been previously published[7,31,32] and are available from the GEO repository with the accession codes GSE42533 and GSE104397. The human breast (https://portal.gdc.cancer.gov/projects/TCGA-BRCA) and lung cancer (https://portal.gdc.cancer.gov/projects/TCGA-LUAD, https://portal.gdc.cancer.gov/projects/TCGA-LUSC) datasets are available from the TCGA website[3,46]. All the other data supporting the findings of this study are available within the article and its supplementary information files and from the corresponding author upon reasonable request. A reporting summary for this article is available as a Supplementary Information file.

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

## Acknowledgements

We thank the members of the Andrechek laboratory for helpful discussions. We thank the Michigan State Investigative HistoPathology Laboratory for the assistance with staining. This work was supported in part by Michigan State University through computational resources provided by the Institute for Cyber-Enabled Research. This work was supported with NIH R01CA160514 and Worldwide Cancer Research WCR - 14-1153 to E.R.A as well as NIH 1F99CA212221–01 to J.P.R.

## Author contributions

J.R. and E.A. collaborated on the study conception, design, and interpretation of results. M.S. provided annotation for translocations and CN analysis. Y.Z. assisted with copy number validation. C.L., E.B., M.J., C.D., and W.H. provided assistance with in vitro

experiments. M.P.O., B.T., and S.Y.L. provided assistance with tail vein injections. B.T. contributed to graphic design of Fig. 1A. C.R., K.S., and K.H. collected samples and performed WES. K.H. assisted in the writing of the manuscript and WES study design. J.R. performed all other experiments and drafted the manuscript. All authors have critically read, edited, and approved the final version of the manuscript.

## Additional information

**Competing interests:** The authors declare no competing interests.

