## [Peer Review File · Nature Communications]

Reviewers' comments:

Reviewer #1 (Remarks to the Author):

This manuscript presents results from whole genome sequencing of two common GEMMs for breast cancer - MMTV-Neu and MMTV-PyMT. The data show that spontaneous genetic changes, including both CNVs and SNVs, can be found at significant frequency in these models, and, importantly, in at least two cases, the targets of mutation are also altered in human cancers. While this is not the first study to suggest that mouse models have value as genetic tools to identify both generic cooperating mutations in cancers, as well as disease-specific interactions, this study is valuable in demonstrating that whole-genome techniques can reveal as yet unappreciated genetic events with biological and clinical significance in human cancer. In one case, amplifications of Col1a1 and CHAD are found to promote metastasis, and this is likely the case in human breast cancer as well - interestingly with a preference for such amplification in ErbB2-driven disease, consistent with the model used here. In a second case, high frequency point mutation of PTPRH was found to likely cooperate with PyMT. While the impact of PTPRH in breast cancer remains unclear, the authors then show that mutation of this EGFR inhibitor occurs in a significant fraction of lung cancers, which are otherwise well understood to be commonly driven by deregulated EGFR activity. Interestingly, PTPRH-mutant cancers are likely to be sensitive to anti-EGFR therapies, and thus this genetic prediction from a GEMM could lead to inclusion of an additional subset of lung cancer patients as good candidates for anti-EGFR therapy.

Overall the genetic data are intriguing and of significance to the field. Particularly in the current era of skepticism of the value of GEMMs as true human cancer models, the demonstration that whole genome analysis can reveal human-relevant oncogenic functions is important to emphasize. Moreover, the identification of the Col1a1 and CHAD functions in HER2 breast cancer and PTPRH mutation in lung cancer are important observations in their own right. To improve the overall impact of the work and better validate the function of Col1a1 and CHAD, however, the following experimental details should be addressed:

1. Figure 2 presents data from clonal cell lines derived from two mouse breast cancer cell lines, and shows that Col1a1 or CHAD KO reduces cell migration and lung seeding. However, it is not clear that clonal control cells were analyzed to assess clonal variation in the parent cell line. Were multiple non-targeting clones derived and tested for equivalent activity to parental cells in these assays? Note that the "rescue" experiments in Figure S3 are not particularly compelling in showing that all of the impairment of these properties is the result of KO, so assessment of variation in control cells is crucial.
2. As with point 1 above, clonal variation in nontargeted BT474 human cell lines should be determined. In addition, as the metastases evaluated in panels G and H are derived from orthotopic injection, it is important to know if these KD/KO cells have deficits in proliferation in culture and/or show impaired primary tumor formation.
3. Based on data shown in Figure S2, BT474 Col1a1 KO1 appears not be a knockout, yet there are phenotypes shown in Figure 3 for this cell line. this needs to be clarified.

Reviewer #2 (Remarks to the Author):

The manuscript by Rennhack et al., examined the genomic profiles of murine mammary cancers driven by MMTV-Neu and MMTV-PyMT. Their results show some consistency with the human counterparts, but importantly, they were able to validate the metastatic effects of Collagen 1 Type 1 alpha 1 (Col1a1) and Chondroadherin (CHAD) both in vitro and in vivo.

This work is the beginning of important reanalysis of GEMMs tumors using the most advanced

genomic and gene editing approaches. My concerns are described below:

1. The genomic analysis still relies on array data which is adequate but not contemporary for gene expression. The genome data is focused on copy number and single nucleotide substitutions and limited analysis of gene mutations, but the presentation is not adequate for a genomics paper. Our interest would be in the genes that are indeed mutated in a genome wide basis. The current presentation cherry-picked the genes of interest. Moreover, the genomic information could identify rearrangements and translocations, which would extend the interest of this manuscript to a wider audience.

2a. The comparisons to human cancers should assess the genome wide transcriptome configurations relative to the common classifications, i.e., luminal A and B, HER2, basal, etc. This analysis was referred to another manuscript but what we would be interested in are the individual cancers in this cohort. Are all MMTV-PyMT tumors basal? Are all MMTV-Neu in the HER-2 positive class, and if not, why not. Figure 1B suggests that the transcriptome of the tumors are not well correlated with the driver oncogenes, nor with histology, but clustered in different expression subsets. The legend states that the clustering is alongside with human cancers, but I cannot discern any annotation of human tumors, rendering this figure uninterpretable. This is a shame since I suspect there is plenty of data buried in figure 1.

2b. The type of analysis using unsupervised clustering is rather basic. Even a PCA analysis would provide better understanding of the fundamental transcriptional structure of these two cancer types.

2c. I find it puzzling that the different drivers could not be discerned in transcriptome analysis. This is more puzzling given the consistency of the genomic data in separating the two driver tumor types. This raises some questions as to the quality of the array analysis and whether batch effects are causing this confusion.

3. One advantage of cross species comparisons of cancer is whether the amplicons found in human cancers (such as the HER2 amplicon) encompasses genes that are also augmented in the GEMMs models. Given the systemic regions, are any genes within the HER2 amplicon in human cancers, also found to be over expressed/amplified in MMTV-Neu tumors. This would be evolutionary convergence of genes needed for HER2 driven mammary/breast cancers.

4. The best work in the manuscript are the descriptions of the functional importance of PTPRH and Col1a1.

Reviewer #1 (Remarks to the Author):

1. Figure 2 presents data from clonal cell lines derived from two mouse breast cancer cell lines, and shows that Col1a1 or CHAD KO reduces cell migration and lung seeding. However, it is not clear that clonal control cells were analyzed to assess clonal variation in the parent cell line. Were multiple non-targeting clones derived and tested for equivalent activity to parental cells in these assays? Note that the "rescue" experiments in Figure S3 are not particularly compelling in showing that all of the impairment of these properties is the result of KO, so assessment of variation in control cells is crucial.

We agree with the reviewer's comment given the light of recent work about the clonal heterogeneity within cell lines (Ben-David, 2018). To address this concern, we have added multiple clonal controls to the *in vitro* wound healing experiments (Figure 4H, bars labeled NDL WT 1/2 and 419 WT 1/2 and, and Figure S7, bars labeled 419-WT-1/2). We have also added a single cell control to the *in vivo* tail vein experiments (Figure 4F, Labeled NDL Clone). These clones closely recapitulated the wildtype findings in the manuscript and were noted in appropriate places in the text. We believe that these findings along with the pooled BT-474 knockout data sufficiently address the concerns about clonal diversity.

2. As with point 1 above, clonal variation in nontargeted BT474 human cell lines should be determined. In addition, as the metastases evaluated in panels G and H are derived from orthotopic injection, it is important to know if these KD/KO cells have deficits in proliferation in culture and/or show impaired primary tumor formation.

We apologize for the confusion with this section. The BT-474 knockdown system was not clonal as with the NDL2-5 and 419 experiments. These experiments leveraged a lentiviral mediated stable transfection of the guide and CRISPRi protein. This population was then selected as a whole, and not single cell sorted as with traditional CRISPR-cas9 editing. We have clarified this in the methods section as well as the relevant sections in the results.

We agree that it is important to note proliferation rate for these cell lines *in vitro* and *in vivo*. There was no significant difference in the growth rate in the wildtype or either Col1a1 or CHAD knockdown in the BT-474 context, so this data was excluded in the original draft of the manuscript. To make this clearer to the reader we have added a supplemental growth rate figure for the *in vivo* tumor growth as well as *in vitro* proliferation (Figure S6). Our knockouts and wildtype tumors show near 100% engraftment rate as noted previously using the technique utilized in the manuscript (Price 1996). We have noted this in the manuscript.

3. Based on data shown in Figure S2, BT474 Coll1a1 KO1 appears not be a knockout, yet there are phenotypes shown in Figure 3 for this cell line. this needs to be clarified.

We believe that the protein produced in Coll1a1 KO1 has taken advantage of alternative start sites present in the Coll1a1 locus. We believe that the band visible on the western blot is a non-functional protein and is still responsible for the phenotype. To confirm this, we added mason's trichrome staining to the manuscript in the knockdown tumors. This staining shows a decrease in collagen fiber production with the loss of Coll1a1 and CHAD. The results have been noted in the manuscript and representative images were added to the relevant figure (Figure S5G)

Reviewer #2 (Remarks to the Author):

The manuscript by Rennhack et al., examined the genomic profiles of murine mammary cancers driven by MMTC-Neu and MMTC PyMT. Their results show some consistency with the human counterparts, but importantly, they were able to validate the metastatic effects of Collagen 1 Type 1 alpha 1 (Coll1a1) and Chondroadherin (CHAD) both in vitro and in vivo.

This work is the beginning of important reanalysis of GEMMs tumors using the most advanced genomic and gene editing approaches. My concerns are described below:

1. The genomic analysis still relies on array data which is adequate but not contemporary for gene expression. The genome data is focused on copy number and single nucleotide substitutions and limited analysis of gene mutations, but the presentation is not adequate for a genomics paper. Our interest would be in the genes that are indeed mutated in a genome wide basis. The current presentation cherry-picked the genes of interest. Moreover, the genomic information could identify rearrangements and translocations, which would extend the interest of this manuscript to a wider audience.

In the original manuscript, the reviewer correctly points out that we identified genes through whole genome sequencing on a relatively small number of tumors then identified the presence of interesting mutations and copy number changes in a much larger cohort of tumors as shown in new figures 4A. Upon the request of the reviewer we have put more emphasis on the genes which were altered in our genomic analysis. We have made the following specific changes as well as altered the discussion to outline the limitations of this analysis.

We have updated the included supplemental tables to include gene names for those genes which have mutations (Table S3), are altered by copy number changes (Table S4), or are affected by a translocation (Table S5). We have also created a new figure (Figure 2) which shows consistently altered genes in each model (Figure 2A). We have also identified gene ontology groups which are consistently altered in each model group (Figure 2B). Furthermore, we have visualized this for the MAPK signaling pathway (Figure 2C). We have also put an emphasis on copy number and translocations throughout the text.

2a. The comparisons to human cancers should assess the genome wide transcriptome configurations relative to the common classifications, i.e., luminal A and B, HER2, basal, etc. This analysis was referred to another manuscript but what we would be interested in are the individual cancers in this cohort. Are all MMTV-PyMT tumors basal? Are all MMTV-Neu in the HER-2 positive class, and if not, why not. Figure 1B suggests that the transcriptome of the tumors are not well correlated with the driver oncogenes, nor with histology, but are clustered in different expression subsets. The legend states that the clustering is alongside with human cancers, but I cannot discern any annotation of human tumors, rendering this figure uninterpretable. This is a shame since I suspect there is plenty of data buried in figure 1.

We apologize for the oversight and confusion. To address this concern, we have added supplemental figure S3 in which we have co-clustered human samples of each molecular cohort with the samples collected in this study. Our findings replicate others (Hollern BCR 2014 and others listed in the manuscript) that mouse models largely cluster together with a portion of the tumors representing various human tumors, which are largely HER2 positive or basal.

The revised figure also makes it easy to address the reviewer's question about whether all MMTV-Neu tumors are in the HER2+ve class. Clearly this is not the case as some Neu tumors cluster with other human subtypes. This is likely due to key differences between the model and human cancer, including a lack of amplification / overexpression in the mouse model of the genes that are co-amplified with HER2 in human breast cancer.

2b. The type of analysis using unsupervised clustering is rather basic. Even a PCA analysis would provide better understanding of the fundamental transcriptional structure of these two cancer types.

We thank the reviewers for this suggested method of visualization of the data. We have included it as a new supplemental figure (Figure S2) and commented on it in the results section. This analysis has come to the same conclusion as our clustering figure in which there is considerable diversity within each model. Interestingly, histology is associated with the greatest diversity in gene expression and is likely a functional readout of the passenger events.

2c. I find it puzzling that the different drivers could not be discerned in transcriptome analysis. This is more puzzling given the consistency of the genomic data in separating the two driver tumor types. This raises some questions as to the quality of the array analysis and whether batch effects are causing this confusion.

Given the similar signaling pathways of the MMTV-PyMT and MMTV-Neu mouse models that have previously been published, we are not that surprised that the transcriptional differences are not readily apparent in the two models (Yeo 2016). This is also compounded by the diversity within the models. We have expanded the discussion section to include this as well as the relevant literature.

During this analysis we have used standard BFRM correction (Carvalho 2008) to remove batch effects between the two cohorts. We have added a supplemental figure (Figure S1) to show the before and after batch effect correction principle component analysis. We have also added a specific section in the materials and methods section focusing on the removal of batch effects. In addition to BFRM, we have also tested COMBAT and found similar results.

3. One advantage of cross species comparisons of cancer is whether the amplicons found in human cancers (such as the HER2 amplicon) encompasses genes that are also augmented in the GEMMs models. Given the systemic regions, are any genes within the HER2 amplicon in human cancers, also found to be over expressed/amplified in MMTV-Neu tumors. This would be evolutionary convergence of genes needed for HER2 driven mammary/breast cancers.

We thank the reviewer for this analysis and have created a new Figure 3 outlining the results of this analysis. We identified a number of times in which the mouse and human amplification or deletion events overlapped but were not entirely conserved (Figure 3A). Specifically, we showed the MMTV-Neu and HER2+ subtype shared a number of genes which have copy number amplification. Surprisingly, a larger portion of genes with alterations were found in the PyMT model and shared with the human disease (Figure 3B). To investigate this for known driving oncogenes and tumor suppressors we showed that two of the MMTV-Neu tumors did contain events that were identified as known drivers in the human HER2+ subtype (Figure 3C).

4. The best work in the manuscript are the descriptions of the functional importance of PTPRH and Coll1a1.

We thank the reviewer for these comments. We think that these events are important and relevant events in human cancer and serve as an important proof of concept analysis as to the importance of studying GEMMs. We have expanded the analysis of this section and believe we have made the findings

more scientifically rigorous and easier to interpret to expand the impact of this paper.

REVIEWERS' COMMENTS:

Reviewer #1 (Remarks to the Author):

The authors have thoroughly addressed my concerns from the initial review. I believe the work as it now stands makes a novel and important contribution to the field in demonstrating genetic and functional similarity between mouse models of breast cancer and human tumors.

Reviewer #2 (Remarks to the Author):

The authors answered my questions and I am comfortable with the current manuscript. I believe this work underscores the importance of murine models of cancer as a discovery tool. The genomic complexity of human cancers and the inability of the human cancer cell lines to recapitulate primary cancer cell biology highlight the importance of GEMMs models to deconvolute this genomic complexity.